# Recent Advances in Wearable Devices for Non-Invasive Sensing

Su Min Yun †, Moohyun Kim †, Yong Won Kwon †, Hyobeom Kim †, Mi Jung Kim †, Young-Geun Park † and Jang-Ung Park *

Nano Science Technology Institute, Department of Materials Science and Engineering, Yonsei University, Seoul 03722, Korea; suminyun@yonsei.ac.kr (S.M.Y.); 2020324081@yonsei.ac.kr (M.K.); kyw0115@yonsei.ac.kr (Y.W.K.); 2019324063@yonsei.ac.kr (H.K.); mjkim85@yonsei.ac.kr (M.J.K.); younggeun@yonsei.ac.kr (Y.-G.P.)
* Correspondence: jang-ung@yonsei.ac.kr
† These authors contributed equally to this work.

**Abstract:** The development of wearable sensors is aimed at enabling continuous real-time health monitoring, which leads to timely and precise diagnosis anytime and anywhere. Unlike conventional wearable sensors that are somewhat bulky, rigid, and planar, research for next-generation wearable sensors has been focused on establishing fully-wearable systems. To attain such excellent wearability while providing accurate and reliable measurements, fabrication strategies should include (1) proper choices of materials and structural designs, (2) constructing efficient wireless power and data transmission systems, and (3) developing highly-integrated sensing systems. Herein, we discuss recent advances in wearable devices for non-invasive sensing, with focuses on materials design, nano/microfabrication, sensors, wireless technologies, and the integration of those.

**Keywords:** wearable sensors; stretchable devices; non-invasive sensing; wireless technologies; smart contact lenses; skin-interfaced sensors

## 1. Introduction

Wearable sensors have attracted significant interest due to their potential ability to provide continuous, real-time health monitoring of various physiological parameters [1–5]. Such ability provides real-time feedbacks on daily diets, activities, and physiological conditions, allowing timely and appropriate control and treatments [6,7]. However, current commercial wearable devices such as smartwatches, wristbands or glasses, are usually just miniaturized forms of conventional electronics that consist of rigid, bulky, and planar components. Such rigidity and bulkiness could not only make the devices obtrusive and disturbing but also restrict the conformal contact of the devices to the human body. In particular, this incomplete contact might result in inaccurate measurements and large noises affected by body movements and environmental conditions [2].

Current research has focused on constructing fully-wearable sensing systems, as depicted in Figure 1. To establish such systems, soft materials, wireless technologies, and high integrity are important factors. There has been significant progress in soft, stretchable, and flexible conducting materials, and this has stimulated the development of wearable devices. Soft conducting materials could enhance the compliance of the device to the human body; therefore, they can not only mitigate the disturbance and obtrusiveness on human skin but also enhance the operational reliability [3,6,8,9]. Wireless power supply and data transmission technologies have also been developed significantly and attempt to incorporate those technologies have been made extensively [10]. Wireless power transfer has advantages in that they only require a simple antenna circuit, which offers a wide selection of materials and structures. Energy harvesters using tribo- and piezoelectric mechanisms enable the charging of electricity through body movement, thus provides

an easy way of charging irrespective of the environmental conditions [11–13]. Progress in wireless communication technologies and electronic engineering facilitates the data processing-transmission procedure with a miniaturized single electronic chip. Among various wireless communication protocols, near-field communication (NFC) and Bluetooth low energy (BLE) have been used most commonly in wearable devices. Their high compatibility with commercial electronic devices, such as smartphones enables easy and simple data recording without any other display devices attached to the sensor [14–16]. Combining all the functional components with varying functions, properties, and shapes into small, compact devices should involve careful and elaborate methodological approaches. The integration should form a configuration suitable for each platform and target function such that they can offer convenient and reliable diagnostics.

Herein, we have discussed the recent progress in wearable sensors for non-invasive sensing. We have summarized recent advances in materials and structural modifications for wearable conductors. Next, we provide an overview of nano/microfabrication techniques that are widely used in the fabrication of wearable devices. Then, we discuss the recent progress in wearable sensors with focuses on their sensing mechanisms and applications. Technologies for wireless energy supply and data transmission are also reviewed. Lastly, we introduce recent studies that demonstrate highly-integrated sensing systems and discuss the challenges and prospects for further development of wearable devices.

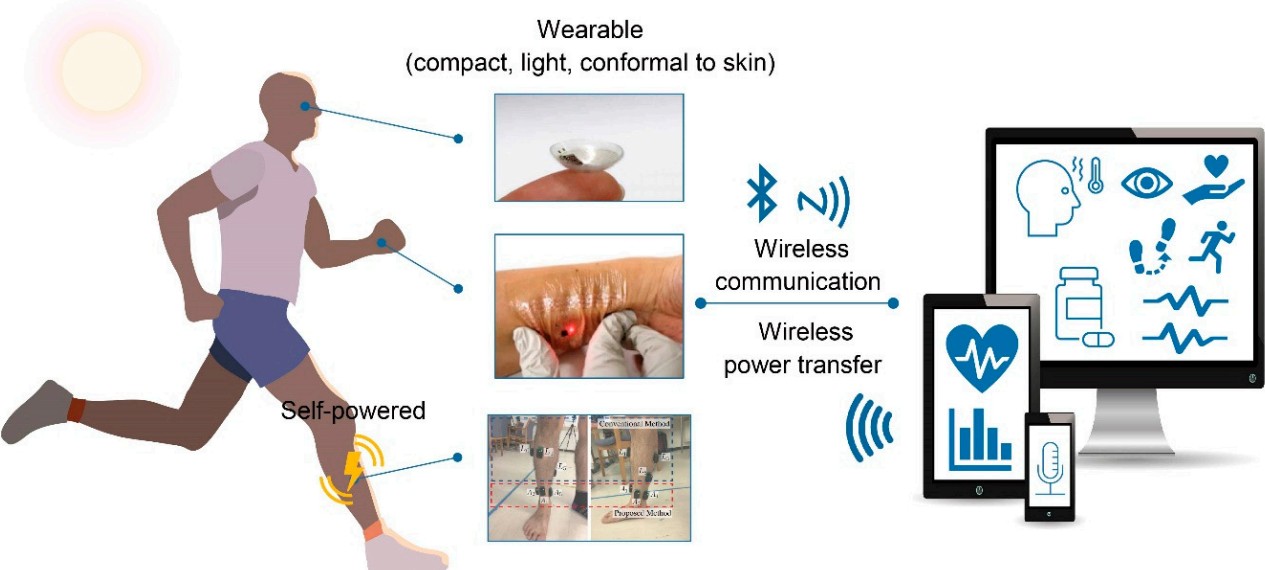

**Figure 1.** Schematics illustrating the recent progress in wearable devices for non-invasive diagnosis. (**top**: reprinted with permission from [14]. Copyright 2020 American Association for the Advancement of Science; **middle**: reprinted with permission from [17]. Copyright 2019 John Wiley and Sons; **bottom**: reprinted with permission from [18]. Copyright 2019 MDPI).

## 2. Materials and Structural Design

### 2.1. Materials

Despite significant progress in wearable devices, electrodes that are incorporated in the devices are usually rigid and planar, thus prone to mechanical and electrical destruction under repetitive stress or strain. Therefore, it is encouraged to develop stretchable and flexible conducting materials that are highly resistant to such stress and strain while providing high conductivity [19,20]. In this section, we summarize the properties and application examples of various conducting materials including metal-based materials, carbon-based materials, conducting polymers, liquid metal, and ionic liquids.

2.1.1. Metal-Based Materials

In this section, we introduce various metallic materials that show promise for use in stretchable and flexible electrodes. By using micro/nanofabrication methods, metallic materials can be deposited or printed on substrates to form electrodes. Inorganic nanomaterials can be synthesized in various dimensions, including zero-dimension (0D, e.g., metal nanoparticles), one-dimension (1D, e.g., metal or metal oxide nanowire/nanofiber), and two-dimension (2D, e.g., transition metal dehydroxylated nanosheet). Nanomaterials of different dimensions can be integrated with other materials to enhance their intrinsic properties. As shown in Figure 2a, 0D materials, such as silver nanoparticles (AgNPs) can be deposited in a flexible and stretchable matrix such as polyurethane to form a percolation network [21]. In this case, the electrical resistance is dictated by the intrinsic electrical conductivity of the nanomaterials and the junction–junction resistance between the individual nanoparticles. More importantly, the percolation network could absorb mechanical strain while ensuring electrical conductivity [22]. Cu-based core/shell NPs were used to fabricate electrodes with uniform electrical properties and high mechanical stability. In particular, the $Cu_{10}Sn_3$ shell with a low melting point facilitated the facile triggering of an efficient densification reaction, thereby resulting in high electrical properties even on a vulnerable polyethylene naphthalate (PEN) substrate [23]. Nanowires (NWs) are 1D materials that are used extensively in conductors due to their high conductivity, simple solution synthesis, and device fabrication process. Nanofibers (NFs) are also a kind of 1D material like NWs, but they have a higher aspect ratio than NWs, which can reduce the number of junction points between the nanomaterials. For example, silver nanofibers (AgNFs) have been used to replace rigid indium tin oxide (ITO)-based film because the AgNF network has good chemical stability, conductivity, transparency, and flexibility. With such characteristics, AgNFs are widely used with transparent or soft substrates such as polyethylene terephthalate (PET) and glass fabrics for the fabrication of flexible devices [24–26]. As shown in Figure 2b, a pressure sensor based on Au NWs identified the subtle difference in pulse, and it may become a wearable diagnostic device for the real-time monitoring of human health under various conditions [27–29]. 2D inorganic material atomic sheets are thin layered crystalline solids with the properties of interlayer van der Waals bonding and intralayer covalent bonding. The excellent properties of 2D crystals generate enormous benefits for traditional semiconductor technology and emerging flexible nanotechnology, especially with their ultimate thickness scalability [30]. For example, research involving 2D molybdenum disulfide ($MoS_2$), which has a similar structure to graphite, has been studied actively. $MoS_2$ easily can be manufactured as a thin film, and it has excellent mechanical, electrical, and optical properties, suggesting the potential for 2D nanomaterials in various wearable sensor applications [31–33]. Choi et al. described a high-density, hemispherical, bending-image sensor array that was designed using strain release devices and thin $MoS_2$-graphene heterostructures at the atomic-level [33].

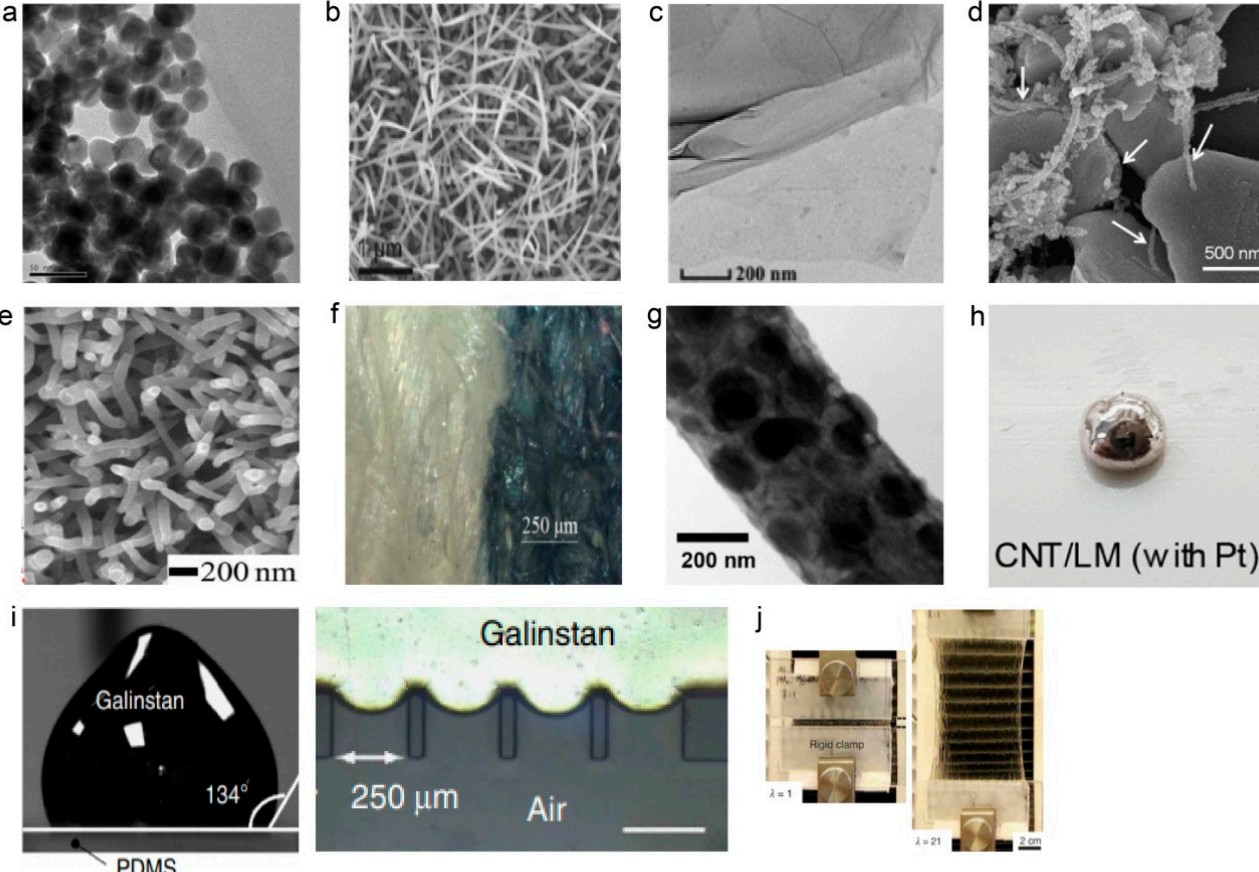

**Figure 2.** (**a**) Transmission electron microscopy (TEM) image of Ag NPs. Scale bar, 50 nm (reprinted with permission from [21]. Copyright 2018 Publish by MDPI); (**b**) Scanning electron microscopy (SEM) image of silver nanowires, Scale bar, 1 um (reprinted with permission from [29]. Copyright 2018 IEEE); (**c**) TEM image of graphene oxide (GO). Scale bar, 200 nm (reprinted with permission with [34]. Copyright 2019 American Chemical Society); (**d**) SEM image of Ag flakes anchored with Ag Nanoparticles/amine-functionalized multi-walled carbon nanotubes (NH$_2$-MWCNT) hybrid material (reprinted with permission from [35]. Copyright 2017 nanoscale); (**e**) SEM image of the PPy nanowires. Scale bar, 200 nm (reprinted with permission from [36]. Copyright 2017 WILEY-VCH Verlag GmbH & Co. KGaA, Weinheim, Germany); (**f**) Optical image of the poly (3,4-ethylenedioxythiophene) polystyrene sulfonate (PEDOT:PSS) electrode on cotton textile. Scale bar, 250 um (reprinted with permission from [37]. Copyright 2019 IEEE); (**g**) TEM image of a poly (vinylidene fluoride-co-hexafluoropropene) (PVDF-HFP)/PEDOT nanofibers (NF). Scale bar, 200 nm (reprinted with permission from [38]. Copyright 2018 Springer Nature); (**h**) Optical image of LM/CNT composite using CNTs decorated with Pt nanoparticles. (reprinted with permission from [39]. Copyright 2019 American chemical society); (**i**) High conductivity and surface tension of Galinstan (left) and optical image of a device after Galinstan injection (right). Scale bar, 250 um (reprinted with permission from [40]. Copyright 2014 Springer Nature); (**j**) Stretchable and tough hydrogels. The stretched to 17 times its initial length (right) (reprinted with permission from [41]. Copyright 2012 Springer Nature).

### 2.1.2. Carbon-Based Materials

Compared with other conductive materials, advanced carbon-based materials, such as carbon black nanoparticles (CBNPs), carbon-based nanofibers, graphene, and carbon nanotubes (CNTs), have unique advantages, including high chemical and thermal stability and good electrical conductivity [42]. It is also easy to functionalize carbon-based materials, which gives them great potential in wearable electronic products and applications [43–45]. Graphene is considered to be a desirable 2D material in the preparation of high strength and multi-functional hydrogels and elastomers due to its high modulus, high strength, large specific surface area, good electrical conductivity, and biocompatibility [46,47]. Graphene oxide (GO) is synthesized by replacing the partial double bonds on the surface of a graphene sheet with oxygen-containing functional groups, such as the hydroxyl group (−OH), the

carboxyl group (−COOH), and the epoxy group (−C−O−C) as shown in Figure 2c [34,48]. Therefore compared with graphene, GO can be dispersed more stably in a variety of organic solvents, thereby providing the conditions for the preparation of various high-strength, multi-functional composites [46]. CNTs have hollow, nano-tubular structures that are obtained by curling single or multi-layer graphene nanoplates. CNTs can be ideal wearable sensor materials due to their excellent electrical and mechanical properties and their availability [49,50]. Lee et al. proposed a scalable strain sensor by coupling overlapped CNT bundles with silicone elastomers. The sensor showed excellent sensing performance with a broad sensing range of over 145% strain, ultrahigh sensitivity (maximum gauge factor of 42,300), high repeatability, and durability [51]. CNT can be used extensively in other wearable healthcare systems in hybridization with various materials. For example, CNT is used as a 3D printable material with high conductivity and low-temperature processability by forming a carbon-metal hybrid combined with Ag nanoparticles, Ag flakes, and a thermoplastic copolymer, as shown in Figure 2d [35]. CNT can be used in other wearable healthcare systems. By mixing single-walled CNT (SWCNT) with a thermally reversible, self-healing polymer, a flexible thermal sensor with mechanical adaptability was developed to monitor the temperature of human skin in real-time [52].

### 2.1.3. Conducting Polymers

Conducting polymers can have flexibility and stretchability that are close to those of traditional plastics, also they have fast carrier transport close to silicon electrons [53]. Poly (3,4-ethylenedioxythiophene) polystyrene sulfonate (PEDOT:PSS) is one of the most extensively studied conducting polymer for flexible and wearable electronics [37,54,55]. PEDOT partially aggregates and forms a "hard" conductive network in the soft PSS matrix due to the weak electrostatic interaction between PSS and PEDOT. Therefore, additives such as STEC enhancers are added to soften the PSS domains, which makes it more applicable in wearable devices. Figure 2g shows conductive core/shell polymer nanofibers that are composed of poly (vinylidene fluoride-co-hexafluoropropene) (PVDF-HFP)/poly(3,4-ethylenedioxythiophene) (PEDOT) [38]. A sponge-like 3D membrane made of nanofibers were fabricated through the electrospinning of PVD-HFP and the following vapor deposition polymerization of PEDOT. The 3D membrane displayed a sheet resistance of approximately $7 \times 104 \, \Omega/\text{sq}$. Synergized with the well-dispersed spherical bumps on the PEDOT shell that lead to the formation of a hierarchical conductive surface, the sponge-like 3D mats exhibited an excellent pressure sensitivity.

### 2.1.4. Liquid Metal Alloys and Ionic Liquids

Liquid metal alloy has good conductivity, fluidity, and flexibility [56]. It occupies the geometry of its container in the liquid phase at room temperature, which prevents mechanical mismatch between the material and the human skin [57,58]. In addition, the liquid metal alloy has exciting functionalities, such as printability, patterning ability, and self-healing [59]. It also can form a 3D interconnect due to the oxide layer that forms spontaneously on its surface [60]. The mechanical and electrical properties of liquid metal can be enhanced significantly by forming a composite with other materials such as metal nanoparticles and CNTs [39]. Eutectic gallium-indium alloy (EGaIn) or a gallium-indium-tin alloy (Galinsta) are two of the most popular liquid metal alloys for wearable devices due to their low viscosity, low toxicity, high conductivity, and high surface tension [40,61]. Guo et al. developed a stretchable wearable EGaIn-based electrocardiogram (ECG) sensor. The EGaIn provided the sensor with high conductivity and stretchability. The electrical connections were kept stable, even when they were stretched over 100% [62]. The ionic liquid is another type of liquid conductor. Unlike liquid metals, ionic liquids can be spread easily onto many elastomers, either forming a uniform coating [63] or becoming trapped between the textile fibers and forming liquid bridges. For example, short-chain ionic liquid species ($[C_2\text{mim}][BF_4]$) formed liquid bridges in small fiber-gaps [64]. The ionic liquid bridge between fibers was used as stretchable conductors and they could respond to the

external mechanical deformation. Such liquid alloy conductive fibers can be used in many fields, including flexible electronic products, smart clothing, interconnects, and antennas.

### 2.1.5. Hydrogels

Hydrogels can form conformal contact with the human body because their mechanical properties are similar to those of the interfaces between human skin and the underlying tissue [41]. Electrolytes and ionic liquids are commonly used to achieve ion transportation in hydrogels [65]. A polyacrylamide hydrogel containing LiCl was used in making ionic interconnects [66]. The hydrogel worked as an ionic conductor and also provided high stretchability and transparency. Cao et al. demonstrated a hydrogel-based aquatic electronic skin that has excellent self-healing properties [67]. The electronic skin consisted of a fluorocarbon elastomer and a fluorine-rich ionic liquid. The ion-dipole interactions between those two materials enabled fast and repeatable electro-mechanical self-healing in wet, acidic, and alkali environments.

### 2.2. *Structural Design*

Despite recent advances in soft materials, wearable devices still have various challenges, such as sensing functionality and long-term reliability, due to the mechanical mismatch with human skin. Considering a significant difference in the mechanical mismatch with the human body, additional structural engineering of devices is needed to achieve conformal attachment and reversible stretching and bending during natural body movements [68–70]. In this section, we overview soft, stretchable devices based on various geometric designs, such as the rigid-soft hybrid, ultrathin, wavy, serpentine, mesh, coil, and sponge designs.

### 2.2.1. Rigid-Soft Hybrid Structure

The rigid-soft hybrid design is a structure in which reinforced islands are embedded in an elastic layer. This design also is referred to as the hybrid design. Each reinforced island and each elastic layer are composed of two materials that have different moduli of elasticity [71]. When deformed, the rigid, reinforced island regions support the active parts of the electronic device without deformation, and the elastic layer resists mechanical deformation by effectively dispersing the strain. Park et al. developed a hybrid substrate composed of rigid, reinforced islands with photo-patternable polymer (SPC-414; $E = \sim360$ MPa) and an elastic layer with silicone elastomer (elastofilcon A; $E = \sim0.09$ MPa), as shown in Figure 3a [15]. For the stretching test, when the external strain was applied (0% and 30%), there was no discernible change in the gap in the interface. This was because the elastic layer of the hybrid substrate provides sufficient stretchability to protect the rigid reinforced regions from mechanical deformation. High transparency (93% in the visible light regime) and low haze (1.6% in the visible light regime) were achieved by using a hybrid substrate, which is suitable for soft, contact lenses.

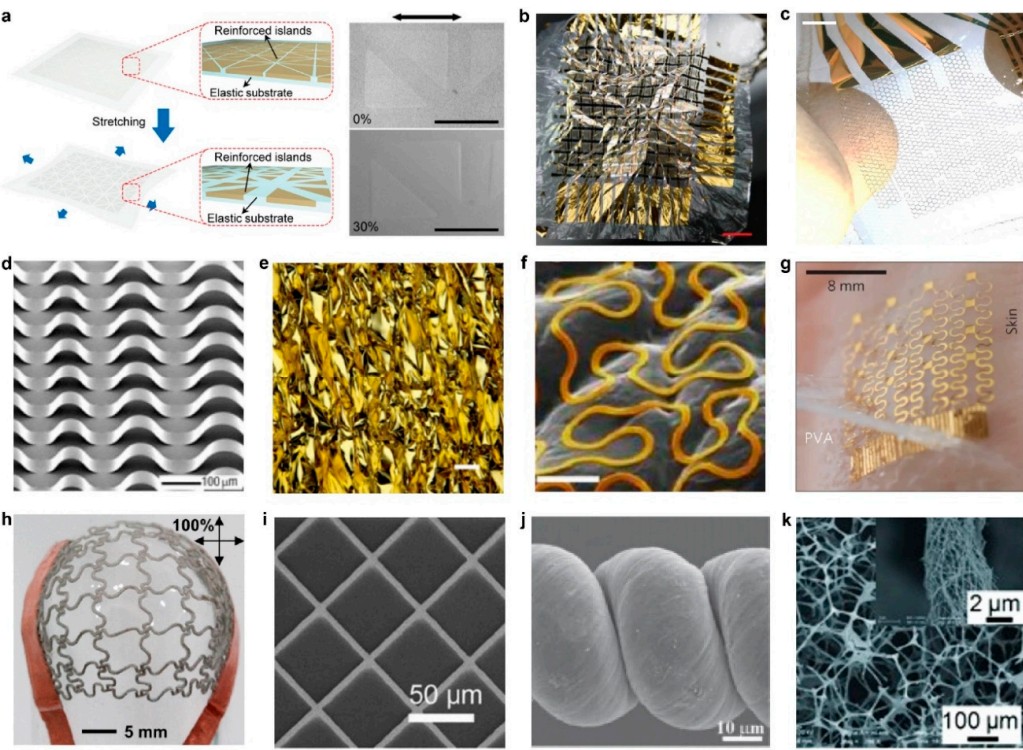

**Figure 3.** (**a**) (Left) Schematic image of the hybrid substrate where the reinforced islands are embedded in the elastic substrate. (Right) SEM images before (top) and during (bottom) 30% stretching. The arrow indicates the direction of stretching direction. Scale bars, 500 µm SEM images before (top) and during (bottom) 30% stretching. The arrow indicates the direction of stretching direction. Scale bars, 500 µm. SEM images before (top) and during (bottom) 30% stretching. The arrow indicates the direction of stretching direction. Scale bars, 500 µm. (reprinted with permission from [15]. Copyright 2018 American Association for the Advancement of Science); (**b**) at only 2 µm thickness, devices are ultraflexible and can be crumpled like a sheet of paper. Scale bar, 1 cm (reprinted with permission from [72]. Copyright 2013 Nature Publishing Group); (**c**) A 4 × 4 stretchable organic electrochemical transistors (OECT) array when intrinsically stretched. Scale bar, 3 mm (reprinted with permission from [73]. Copyright 2018 American Association for the Advancement of Science); (**d**) Si ribbon structures formed on a PDMS substrate pre-strained to 50% and patterned with $W_{act} = 15$ µm and $W_{in} = 250$, 300, and 350 µm (from left to right). The image was taken by tilting the sample at an angle of 45° (reprinted with permission from [74]. Copyright 2006 Nature Publishing Group); (**e**) Corresponding optical micrographs of the wrinkled Au film electrode at 0%. Scale bar, 100 µm (reprinted with permission from [75]. Copyright 2018 American Chemical Society); (**f**) scanning electron microscopy images of third-order Peano-based wires on skin and a skin-replica (colorized metal wires), showing the conformal contact of the wires on the substrate. Scale bar in (**f**), 500 µm (reprinted with permission from [76]. Copyright 2014 Nature Publishing Group); (**g**) Image of a 4 × 4 TCR sensor array after application to the skin using a water-soluble adhesive tape based on poly(vinyl alcohol) (reprinted with permission from [77]. Copyright 2013 Nature Publishing Group); (**h**) Optical image of a final mesh heater after biaxial stretching up to ~100% (reprinted with permission from [78]. Copyright 2015 American Chemical Society); (**i**) SEM images of Ag mesh, demonstrating a space length among Ag lines of 45 µm (reprinted with permission from [79]. Copyright 2018 WILEY-VCH Verlag GmbH & Co. KGaA, Weinheim, Germany). (**j**) High-magnification SEM image of the CNT loops (reprinted with permission from [80]. Copyright 2012 WILEY-VCH Verlag GmbH & Co. KGaA, Weinheim, Germany) (**k**) SEM images of PUS-AgNW-PDMS composites (reprinted with permission from [81]. Copyright 2013 WILEY-VCH Verlag GmbH & Co. KGaA, Weinheim, Germany).

### 2.2.2. Ultrathin Structure

Considering that the induced strains are linearly proportional to the overall thickness, flexible electronics can be achieved by reducing the thickness; therefore the induced strains are decreased [82,83]. Kaltenbrunner et al. introduced plastic electronics with nanometers-thick organic transistors on 1-µm-thick ultrathin polymer (polyethylene naphthalate (PEN)), as shown in Figure 3b [72]. This ultrathin polymer can be crumpled and stretched up to 230%, which enables conformal attachment to arbitrary and curvilinear surfaces, such

as human skin. Lee et al. developed active organic electrochemical transistors (OECTs), which were encapsulated with a 1.2-μm-thick parylene layer, as shown in Figure 3c [73]. Fabricating electronic devices with a total thickness of 2.6 μm enabled the devices to have high flexibility and stretchability. Based on these properties, a cyclic test was conducted by applying a strain of 15%. After the cyclic test, the transconductance and drain current were measured and only had a 7% difference, which showed sufficient mechanical stability.

### 2.2.3. Wavy and Serpentine Structure

To achieve both flexibility and stretchability, wavy and serpentine geometries provide additional benefits through the elongation and contraction of the structure. A device attached to a uniaxial or a biaxial pre-strained elastomer enables the fabrication of a wavy structural device by releasing the strain [84]. Sun et al. introduced a stretchable, periodic, wavy, structural electronic device with an inorganic semiconductor and silicone nanoribbons on a pre-strained polydimethylsiloxane (PDMS) substrate, as shown in Figure 2d [74]. When the PDMS and Si ribbons that were patterned with the activated surface were exposed to UV, -OH is formed on the surface, and a strong siloxane linkage was formed through the condensation reaction. These adhesion regions enable well-defined, wavy geometry to be obtained when the strain is released. The resulting device has high stretchability (~100%), compressibility (~25%), and bendability (curvature radius down to ~5 mm). Nur et al. introduced a highly-sensitive, ultrathin, wrinkled Au film strain sensor [75]. Figure 3e shows an image of the strain sensor with Au/parylene film on a pre-strained dielectric layer. This capacitive-based strain sensor achieved a 3.05 gauge factor and high stretchability to 140% with mechanical stability. A Serpentine design is a certain curved shape with periodically repeating geometry. Such a geometry enables large scale stretchability by releasing strain energy when mechanically deformed and conformally attached to the skin [85]. Fan et al. introduced stretchable electronics with fractal-shaped electrodes patterned on the elastomer, and the devices had a mechanical stretchability of 32% on the *x*-axis and 28% on the *y*-axis, as shown in Figure 3f [76]. This serpentine structure supports conformal attachment to the skin and minimizes the electromagnetic disturbances during magnetic resonance imaging. Webb et al. developed 16 temperature sensors with serpentine geometry interconnect on the microperforated elastomer, as shown in Figure 3g [77]. This geometry minimizes the difference in the strains between the sensors and the actuators, and under an external strain of 10%, the temperature coefficient of resistance (TCR) of the device's strain of sensors/actuators was <0.02%.

### 2.2.4. Mesh Structure

A mesh design is a geometry that also allows a device to attain high stretchability. When an external strain is applied, each empty hole and mesh line plays a role in enabling stretching and bending. Choi et al. introduced a soft and stretchable heater by patterning mesh design nanocomposite of silver nanowires on a thermoplastic elastomer, as shown in Figure 3h [78]. This mesh design enabled conformal attachment with high stretchability (up to ~100%) and uniform heat transfer during the natural movement of the body, allowing it to be used to perform articular thermotherapy. Jiang et al. introduced Ag mesh design electrodes that had high conductivity, stretchability, and mechanical durability, as shown in Figure 3i [79]. This mesh design achieved a conductance of 17 $\Omega$ sq$^{-1}$ R$_{sh}$ at 93.2% transmittance and only 16.6% and 10.6% increase in R$_{sh}$ under 50% and 100% external strain, respectively.

### 2.2.5. Three-Dimensional (3D) Structure

3D structure designs, such as 3D coiled structures and 3D sponge structures, have many advantages. In the case of a 3D coiled structure, deformation of the coil alleviates the maximum strain of the coil, enabling it to be more stretchable than the 2D serpentine structure [86]. Shang et al. presented a yarn-derived, spring-like, long CNT rope that can produce tensile strains of 285% during external elongation by opening the loops, as shown

in Figure 3j [80]. This coiled structure allows 20 times higher stretchability than the conventional pure CNT yarn structure, and it does so without strength and conductivity problems. In the case of the 3D sponge structure, the interconnected 3D porous scaffold structure maintains stretchability and conductivity through shape deformation. Ge et al. introduced binary-network-structured polyurethane sponge-Ag nanowire-poly(dimethylsiloxane) (PUS-AgNW-PDMS) conductors, as shown in Figure 3k [81]. The key role of the PUS as a skeleton to support AgNW networks with a porous and junction-free structure. This structure accommodates the deformation without changes in conductivity under stretching conditions. The result shows high elongation (140%) and high electrical conductivity (19.2 $S\,cm^{-1}$) under an external strain of 50%.

## 3. Nano/Microfabrication Techniques

Various nano/microfabrication techniques including physical vapor deposition (PVD), chemical vapor deposition (CVD), electrodeposition, lithography, and printing have been extensively studied and used for the fabrication of wearable devices. We recommend the readers refer to the following review papers to find more detailed descriptions on the processes and properties of the techniques [87–91].

PVD including thermal evaporation, electron-beam (e-beam) evaporation, and sputtering is widely used for the deposition of metal and metal oxide-based materials into thin films on substrates [92–95]. Polymer solutions and dispersions of various conducting materials such as metal particles, metal wires, or graphene can be deposited onto diverse matrices and substrates using techniques such as spin-coating, vacuum filtration, and spraying [96–99]. The adhesion between the deposited materials and the matrices is achieved by chemisorption or physisorption, and to enhance the adhesion and compatibility between them, surface treatments including $O_2$ plasma and silane coupling reaction are widely exploited [100,101]. Electrodeposition is a process of forming thin films of metal-based materials or polymers by applying electric current onto a conductive material that is immersed in a solution containing precursors such as metal salts, monomeric or polymeric materials [102–104]. Processing temperatures, chemical resistances of substrates and the deposited materials, the required thickness and the uniformity of the deposition, and the geometrical properties of the substrate matrices are various factors to consider for the selection of deposition techniques. For instance, many flexible or stretchable substrates such as PDMS are susceptible to thermal expansion and contraction under temperature variations, which may cause damages on the deposited metal layers, thus favoring low-temperature processes [105].

For the patterning of deposited layers, photolithography that consists of UV exposure, wet etching or dry etching steps, and e-beam lithography are the most widely used techniques [71,106,107]. These techniques allow high-resolution patterning with nm scale feature sizes for e-beam lithography and tens of nm to μm scale feature sizes for photolithography but have disadvantages of high-cost and small patterning area [90]. E-beam lithography provides low throughput compared with other fabrication techniques. Printing techniques such as screen printing, roll-to-roll printing, and inkjet printing are also extensively used for the nano/microfabrication of flexible and stretchable devices [108–111]. In screen printing, inks are deposited and spread on to the mask that is placed and aligned onto a substrate, thus resulting in a selective and templated deposition of the ink onto the desired sites of the substrates [112,113]. The screen printing technique is compatible with diverse matrices including textiles, elastomers, and paper, and the minimum feature sizes that can be obtained using the technique are usually tens of μm, but highly dependent upon the composition of ink and particle sizes [114–116]. Direct writing and 3D printing enable the patterning of various materials onto substrates of diverse geometries and compositions [117–119]. Park et al. demonstrated high-resolution and reconfigurable 3D printing of liquid metal using direct printing as shown in Figure 4i [60]. The resolution of printing was controlled by the diameter of nozzles, and the minimum line width of 1.9 μm was obtained reliably. The printed liquid metal could be lifted-up and relocated by the

nozzle tip to form 3D structures. Soft lithography uses elastomeric molds and stamps that are patterned and engraved into desired patterns to produce patterns of various materials as shown in Figure 4h, which allows relatively simple and quick replication, large pattern areas, and low-cost processing [120,121].

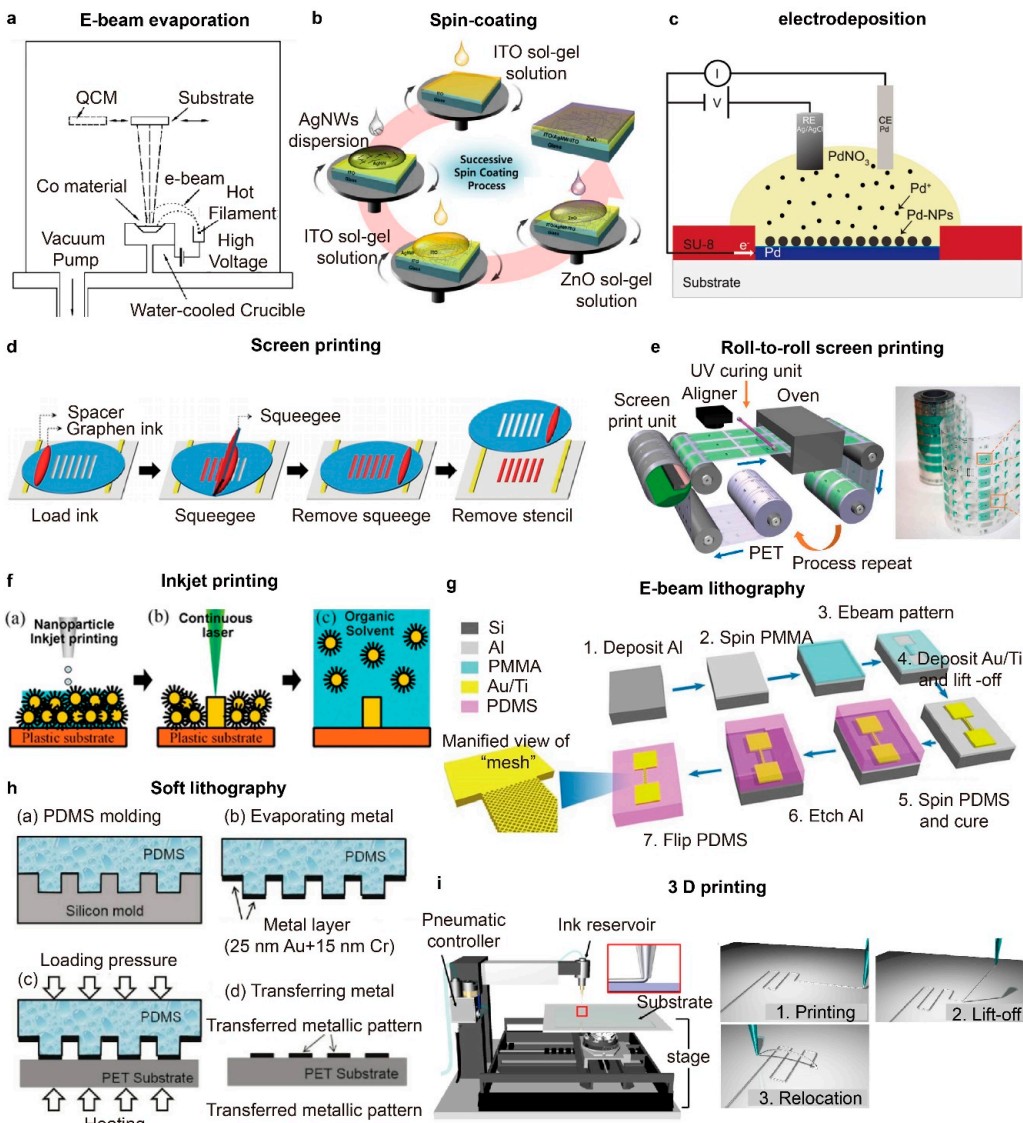

**Figure 4.** (**a**) Schematic Illustration of an e-beam evaporation system (reprinted with permission from [95]. Copyright 2004 American Vacuum Society); (**b**) Schematic illustrations of the successive spin coating ofITO, AgNWs, and ZnO dispersions (reprinted with permission from [99]. Copyright 2015 Royal Society of Chemistry); (**c**) Schematic illustration showing the electrodeposition of palladium nanoparticles (PdNPs) (reprinted with permission from [102]. Copyright 2019 John Wiley and Sons); (**d**) Schematic illustrations showing the process of screen printing using a silicon stencil and a pristine graphene ink (reprinted with permission from [113]. Copyright 2015 John Wiley and Sons); (**e**) (Left) Schematic illustrations of roll-to-roll rotary screen printing of electrodes for sweat sensing patches and (right) an optical image of the as-fabricated sensing electrode patterns (reprinted with permission from [112]. Copyright 2019 The American Association for the Advancement of Science); (**f**) Schematic illustrations of inkjet printing and patterning of AgNPs using laser sintering (reprinted with permission from [108]. Copyright 2017 John Wiley and Sons); (**g**) Schematic illustrations demonstrating the fabrication process of high-resolution gold patterns onto elastomer using spin-coating, e-beam lithography and etching (reprinted with permission from [106]. Copyright 2018 Springer Nature); (**h**) Schematic illustrations of soft lithography involving PMDS molding, metal deposition and metal pattern transfer (reprinted with permission from [121]. Copyright 2017 MDPI); (**i**) (Left) Schematic illustrations of the 3D direct printing system and (right) printing, lift-up and relocation steps (reprinted with permission from [60]. Copyright 2019 The American Association for the Advancement of Science).

## 4. Sensors

Various physical properties of the body can be recorded through the different physical variables presented in the external organs of the body. Highly flexible and stretchable bioelectronics that can have conformal contact with the soft, dynamic, and deformable portions of the body are used as wearable sensors for multi-purpose diagnoses of the human body. These sensors have one or more sensing components that can receive various signals of physical, electrophysiological, and biochemical significance for the long-term monitoring of various health conditions. These sensors are devised to be region-specific, and they experience little interference due to the movement of the body.

### 4.1. Physical Sensors

One of the fundamental conventional parameters of the body that have strong diagnostic significance are physical parameters such as movement-induced strain and temperature. These physical parameters vary temporally and spatially, hence the issues of specificity and accuracy are important when devising such physical sensors. In addition, such parameters are recorded primarily from the skin or they can be incorporated into clothing and artificial epidermal substrate for sensing purposes. Other research has focused on bodily kinetics and muscle activity to study the effect of various diseases that affect the movement of the body, such as cerebral palsy.

The promising aspect of physical sensors is that the devices have stretchable electronics that can detect human motions and perform surface strain calculations. These devices can be incorporated intrinsically into clothing or directly onto the surface of the skin to detect the strain derived from bodily motion. Yamada et al. demonstrated a wearable sensor of single-walled carbon nanotubes that was capable of measuring strains up to 280% [122]. Figure 5a show a photograph of the actual device on synthetic clothes. This sensor can differentiate various aspects of human motion, such as movement, breathing, speech, and typing. This paves the way for an additional wide range of wearable electronics for non-invasive physical sensors. Another work that demonstrates the potential of detecting bodily movement using a wearable sensor was presented by Wang et al. using an ionic liquid microband, as shown in Figure 5b [123]. This intrinsically-stretchable pressure sensor uses nanomaterials and a flexible semiconductor to fabricate a rubber band-like stretchable pressure sensor that is light and thermally sensitive. The use of a non-volatile material also reduces the complexity of the fabrication, and the device showed the detection of strains in the range of 0.1% to 500%, even during multiple repeating cycles. This device can be integrated seamlessly into woven commercial bracelets for the detection of hand gestures, and it even can identify wrist pulses. Work to develop epidermal electronics for a non-invasive physical sensor is a novel component of non-invasive wearable electronics. Tattoo electronics are key devices of such classes due to the seamless infrastructure when they are integrated for application. The work reported by Kim et al. introduced a conformal, contact-based system that shows the great applicability of epidermal tattoo electronics [124]. When incorporated with various sensor systems, the epidermal device can detect various physical variables of the skin, such as temperature and strain.

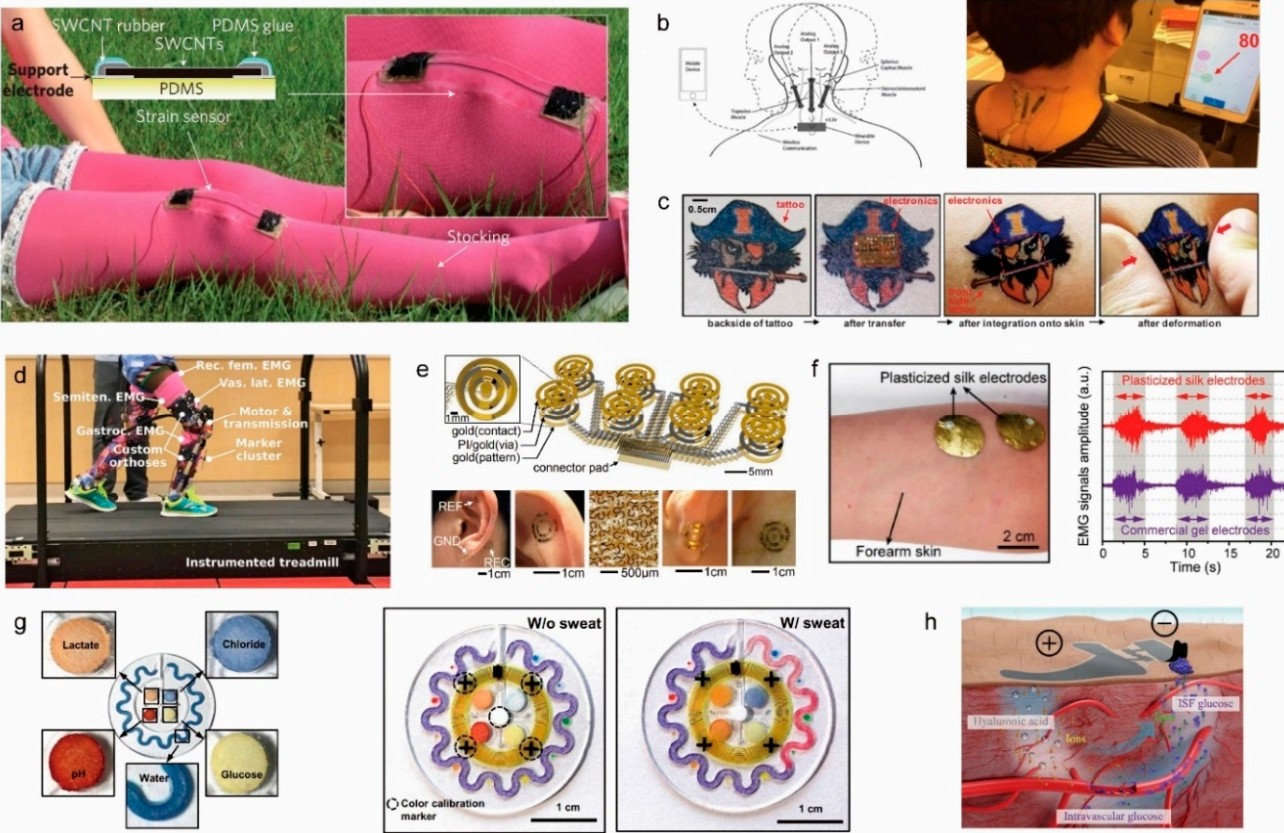

**Figure 5.** (**a**) A photograph of stretchable carbon nanotube strain sensor attached to clothing to detect physical motion (reprinted with permission from [122]. Copyright 2011 Springer Nature); (**b**) Ionic liquid-based sensor patch of cervical movement detection for directional strain calculation of skin during neck movement (reprinted with permission from [123]. Copyright 2014 Royal Society of Chemistry); (**c**) Commercial epidermal tattoo with the integrated circuit at its backside for multifunctional application (reprinted with permission from [124]. Copyright 2011 American Association for the Advancement of Science); (**d**) Photo of the patient using exoskeleton assisted knee extension on an instrumented treadmill (reprinted with permission from [125]. Copyright 2017 Spring Nature); (**e**) (Top) EEG electrode with a tripolar concentric ring with an integrated capacitive design. (Bottom) Images of ring electrodes on the skin for EEG measurement (reprinted with permission from [126]. Copyright 2015 National Academy of Sciences); (**f**) (Left) A photograph of plasticized silk electrode on human forearm for EMG measurement. (Right) EMG signals obtained by silk electrode during muscle movement with comparison to commercial gem electrodes (reprinted with permission from [127]. Copyright 2018 John Wiley and Sons); (**g**) (Left) Colorimetric detection reservoirs for water lactate, glucose, creatinine, pH, and chloride ions in a wearable microfluidic device. (Right) Images of epidermal microfluidic electrochemical sensor after injection of artificial sweat (reprinted with permission from [128]. Copyright 2016 The American Association for the Advancement of Science); (**h**) Glucose detecting mechanism schematics of a skin-like sensor using an electrochemical twin channel (reprinted with permission from [129]. Copyright 2017 The American Association for the Advancement of Science).

To reinforce the concept of the physical sensor, work to integrate physical sensors for the training and rehabilitation of muscles also shows signs of increasing importance. Fiber-shaped strain sensors can be integrated into woven textile materials for real-time mechanical feedback of surface strain detection. These stretchable sensors can be used to monitor sports activity when attached to an athlete's body [130]. Work to improve the usability of these surface strain sensors was developed further by integrating them into the exoskeleton. The knee extension-assisted exoskeleton developed by Lerner et al. shows that the device can help rehabilitate patients with cerebral palsy and weakened muscle activity [125]. With the onset of cerebral palsy commonly occurring during childhood, the importance of addressing the implication of such disease comes early. The work that has been presented shows evidence of improvement in the dynamic gait posture and the

rehabilitative impact of limb joint mechanics in patients using the robotic exoskeleton. The demonstration of the actual rehabilitation process using this device is shown in Figure 5d. The deep mechanical analysis of patient movement during the rehabilitation process also shows promising insight into the study and clinical modeling of cerebral palsy. With these mechanical physical sensors, the long-term investigation of disease and mechanical rehabilitation can be conducted non-invasively.

### 4.2. Electrophysiological Sensors

Another variety of non-invasive sensors are the electrophysiological sensors that collect bodily electrical signals, such as the ECG [131,132], electroencephalogram (EEG) [133,134], and electromyography (EMG) [135,136]. The capability of continuous physiological monitoring is needed for the treatment and diagnosis of medical conditions. The issue related to the acquisition, filtering, amplification, and radio frequency (RF) transmission of data must be optimized for precise electrophysiological characterization. Hence, epidermal sensors have been developed that can seamlessly access the physiological data of the body. These sensors usually are composed of a soft, stretchable substrate with a biocompatible electrode that has a curvilinear structure.

The study of the bioelectrical signals of the body has been extremely important in the early diagnosis of disease and its symptoms. In the case of cardiovascular diseases, the abrupt and random occurrences of heart attacks may require constant recording and monitoring of electrophysiological signals. Most commercial devices require hospitalization and the use of an invasive diagnostics device that can cause secondary complications. Lee et al. developed a thin electronic device that, when placed on the epidermis, is capable of self-adhesion [137]. This device is integrated with commercial electronic components and uses carbon nanotubes to retain conformal contact with the wrinkles of the skin without any motion artifacts. The use of carbon nanotubes has resulted in a drastic lowering of the overall modulus of the device. This enhanced flexibility of the device enables the collection of reliable ECG data, and it has an overall performance that is superior to that of the rigid, capacitive-type electrode. Xu et al. also demonstrated the use of a low-modulus, elastic, epidermal sensor for recording ECGs [138]. They used controlled mechanical buckling and soft microfluidics to fabricate ultra-low modulus devices that are intrinsically stretchable. They also demonstrated the concept of collecting data wirelessly, which is very important for the inconspicuous operation of epidermal electrophysiological sensors.

Recent advances in non-invasive EEG devices have shown long-term operation and minimal loading contact. EEG devices, in particular, are mounted onto the auricle and mastoid surface, which requires minimal thermal, mechanical, and electrical damage to the skin. In addition, the high irregularity of the surface of the auricle requires detailed structural planning for the bending and stretching of these EEG devices. The non-invasive epidermal EEG developed by Norton et al. has a wearable EEG sensor that remains mounted conformally during vigorous exercise [126]. Figure 5e shows the conformal attachment of the device to the skin when the device is mounted. The device has shown the features of non-invasive interfaces to the skin that show reliable EEG recording and the possibility of wireless integration. Admittedly, there is room for the amplification of the signal and improved accuracy since EEG signals usually are small and difficult to trace. To detect detection the electrical activity produced by a skeletal muscle, EMG signals usually are recorded on the surface of the skin directly above the targeted muscle. Chen et al. showed an example of an EMG skin device that used silk protein substrate for a soft and stretchable sensor device that could be integrated seamlessly, as shown in Figure 5f [127]. The device had high electrical performance and conformality when it was used as an epidermal electronic sensor.

### 4.3. Biochemical Sensors

Another category of the non-invasive sensor is the chemical analysis of ions, biomolecules, electrolytes, and proteins present in bodily fluids. The modern clinical examination uses

blood samples for biochemical assessment for various diseases and conditions. But for non-invasive sensors, sweat and its ionic composition are the main targets for quantitative biochemical analysis. This almost completely removes the need for the invasive collection of samples that can cause psychological trauma or problematic infections. Such electrochemical sensors usually are composed of soft, stretchable, microfluidic systems with specialized receptors that interact with the target ions. These receptors are usually enzymes or antibodies that have a specific color or a change in pH when they react with the target ions.

The characterization and biochemical analysis of biofluids, such as sweat, can be deemed to have high clinical value when complemented with other physical characteristics. Due to the relative ease of non-invasive sample collection, sweat is a representative biofluid that is full of intricate biomarkers. Koh et al. demonstrated a soft, wearable microfluidic device that was capable of capturing and storing sweat for electrochemical analysis [128]. The device, which is shown in Figure 5g, allows the quantitative collection of sweat rates, pH variations, and changes in the concentrations of biomarkers. The changes in the biomarkers are analyzed by colorimetry, and the test is conducted during a controlled fitness cycle. The epidermal microfluidic device can evaluate the performances of athletes while monitoring various other markers of health status. Key markers, such as creatinine, lactate, chloride, glucose, and pH, are evaluated without the device losing its adhesion to failing to operate correctly. Another perspiration-based biochemical sensor was developed by Emaminehad et al., and it evaluates the variables related to cystic fibrosis for diagnostic purposes [139]. The electrolyte contents of the sweat of cystic fibrosis patients were obtained through non-invasive methods using sweat collected by the sensor. The device is composed of a miniaturized iontophoresis interface that delivers stimulating agonist to sweat glands via an electrical current. The correlation between the concentration of the drug and the rate of sweating has been demonstrated for clinical modeling with reliable stability of the device and user conformality. It also introduced a preliminary study of metabolic glucose content between sweat and blood. The culmination of the sweat profiles of individuals can enable future integration into large-scale clinical investigations for more extensive biomedical modeling. Another notable work that utilizes sweat for non-invasive recording is the textile-based potentiometric sensor developed by Parrilla et al. for the analysis of multiple ions in sweat [140]. The incorporation of a polyurethane-based, ion-selective membrane with a stretchable ink electrode strengthens the conductivity and biocompatibility of the textile-based sensors. The integration of stretchable ink and components with serpentine design has greatly enhanced the durability of tensile stress. Despite the practicality of sweat-based sensors, there are issues related to the uncertainties associated with the device due to the presence of other biofluids and markers because they limit the impact of the biomedical applications for non-invasive sampling.

To supplement the mentioned uncertainty associated with monitoring sweat, Chen et al. developed a skin-like, non-invasive sensor for the highly-accurate monitoring of glucose in blood [129]. The device has integrated subcutaneous channels that drive intravascular blood glucose to the surface of the skin. The epidermal glucose detecting mechanism is shown in Figure 5h. This additional augmentation does not comprise the conformality of the device, and it provides prospects for high standard clinical application. When the reliability of the device is optimized to the medical-grade standard, the potential possibility of insulin therapy will be opened with further expansion of the application prospective of non-invasive electrochemical sensors.

## 5. Wireless Technologies

A wireless power supply and data transmission are essential techniques for a fully-wearable system. These wireless technologies enable continuous, on-demand monitoring of health without bulky, wired instruments, and this further facilitates the real-time reporting of the obtained data to clinicians, thereby allowing timely and convenient diagnoses.

### 5.1. Self-Powered System

The source of power is one of the key factors for the long-term, stable operation of wearable sensors. Lithium-ion batteries are used most commonly in wearable electronics. However, their bulkiness and rigidity could limit the wearability of the whole integrated system despite other components being flexible and stretchable. To address this issue, various approaches have been used to implement structural modifications, such as the island-bridge, serpentine structures, and origami folding [71,141]. However, lithium-ion batteries still have toxicity issues, and they require intermittent recharging or replacement, which limits the sustainable use of the devices that use them. Therefore, there has been significant interests in developing self-powered systems and using them in wearable electronics. Wireless power transfer and energy harvesting technologies are the two main technologies that are being used in self-powered systems.

#### 5.1.1. Wireless Power Transfer

Electrical energy can be transferred wirelessly through inductive coupling between the antennas of a transmitter and a receiver [142]. In this system, the transmitting antenna converts the alternating current passing through it into oscillating magnetic fields, then, as the magnetic fields reach the adjacent receiving antenna, an alternating current is induced in the receiver device. The amount of electric current that is generated depends on the intensities of the magnetic fields reaching the receiving antenna, so, to achieve sufficient energy transfer, the system requires that the two antennas be located very close to each other. Resonant inductive coupling can be used to increase the distance over which sufficient power can be transferred. Resonant inductive coupling refers to the phenomenon in which the inductive coupling becomes stronger when the resonant frequencies of the two antennas are in the same predetermined range [143]. Through resonant inductive coupling, the efficiency of energy transfer can be enhanced significantly, but elaborate tunning of the resonant circuit is required to match the frequencies of the antenna circuits.

Kim et al. demonstrated a smart contact lens for monitoring glucose and intraocular pressure, in which the power supply and data transmission are acquired wirelessly through the inductive coupling between the antenna embedded into the lens and an external reader coil connected to a network analyzer, as shown in Figure 6a [144]. The reader coil was placed 10 mm away from the smart contact lens to inductively couple and power it. Glucose levels and intraocular pressure were measured simultaneously but independently by the variation in reflection coefficient and the shift of resonance frequency from the data recorded on the network analyzer, respectively. Then, Kim et al. coupled a similar antenna circuit with a rectifier that was comprised of Si diodes and a capacitor to convert the wirelessly-received alternating current (AC) power to direct current (DC) power, as shown in Figure 6b [15]. Then, the DC power turned on the glucose sensor and the light-emitting diode (LED) pixel on the lens to provide a visualized sensing signal. In addition, the spiral antenna and the interconnect were made of Ag nanofibers to provide high transparency and stretchability. Later, to enable the continuous operation of a smart contact lens powered by inductive coupling, Kim et al. incorporated a rechargeable solid-state supercapacitor into a wireless energy transfer circuit, as shown in Figure 6c [145]. The power that was received wirelessly was stored in the supercapacitor. The supercapacitor was made of carbon electrodes and a solid-state polymer electrolyte, and it was formed in an arc-shape on the smart contact lens using a computer-assisted, direct ink writing (DIW) technique. The arc-shaped form provided the efficient storage of energy within the limited area of the contact lens, and it provided durable operation even when being stretched. After 45 s of wireless charging, the supercapacitor turned on the LED pixel placed on the contact lens and it stayed on for more than 60 s. Also, the capacitor showed a negligible change in its charging/discharging behavior, and it retained more than 90% of its original capacity after 30 cycles of charging/discharging, thereby indicating its long-term reliability and durability.

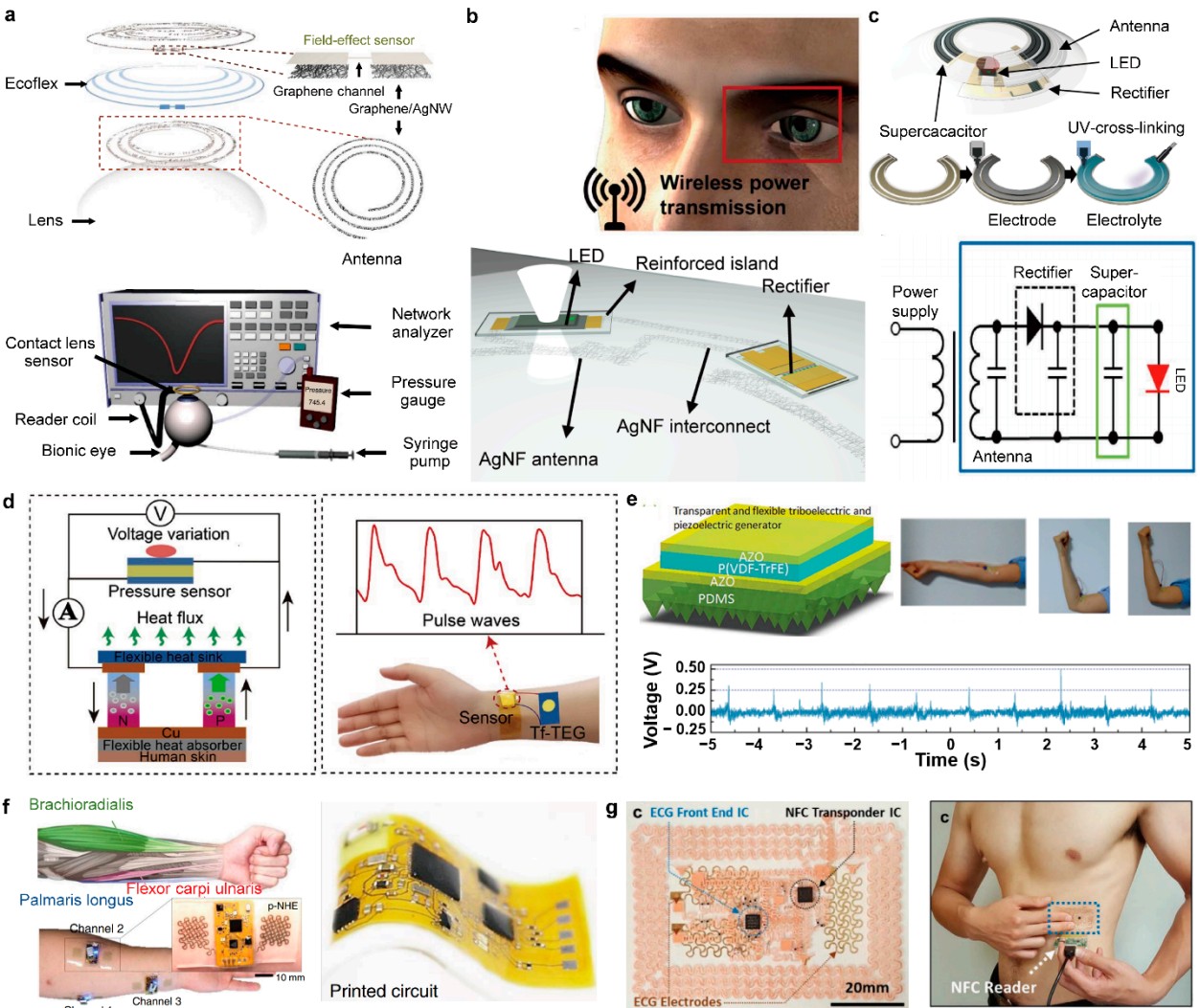

**Figure 6.** (**a**) Wireless power and data transfer between a smart contact lens and a network analyzer via inductive coupling (reprinted with permission from [144]. Copyright 2017 Springer Nature); (**b**) Smart contact lens with a wireless power transfer circuit and a LED display (reprinted with permission from [15]. Copyright 2018 American Association for the Advanced Science); (**c**) Smart contact lens integrated with a wireless power transfer circuit, LED pixel, and printed solid-state supercapacitors for continuous operation (reprinted with permission from [145]. Copyright 2019 American Association for the Advanced Science); (**d**) Self-powered wearable patch for continuous monitoring of pressure driven by a flexible thin-film thermoelectric generator (reprinted with permission from [146]. Copyright 2020 Elsevier B.V); (**e**) Triboelectric and piezoelectric generator with body movement sensing (reprinted with permission from [147]. Copyright 2017 Royal Society of Chemistry); (**f**) Wireless bioelectronics for multimodal human-machine interfaces via BLE (reprinted with permission from [148]. Copyright 2020 Springer Nature); (**g**) E-tattoos for wireless, personalized multimodal sensing via NFC (reprinted with permission from [17]. Copyright 2019 John Wiley and Sons).

### 5.1.2. Energy Harvesting

Energy harvesters convert non-electrical energy into electricity, which means that they can generate power sustainably as long as the non-electrical sources are supplied. Photovoltaic cells, biofuel cells, and thermoelectric nanogenerators generate electricity from light, biomolecules, and heat, respectively. Triboelectric and piezoelectric nanogenerators convert mechanical forces into electricity, hence, they have an advantage in that energy sources can be acquired easily and continuously from body movement irrespective of the environmental conditions. In addition, since biofuel cells and thermo-, tribo-, and piezo-electric nanogenerators can generate electricity in proportion to the magnitudes of the non-electrical sources they convert, they also can operate as biosensors [149–151]. In wearable

systems, energy harvesters should form a conformal contact to human skin such that energy sources from the body can be transferred efficiently to the devices. Therefore, the devices need to possess high softness, flexibility, and stretchability. Wang et al. combined a flexible, thin-film thermoelectric generator (tf-TEG) with a pressure sensor to build a self-powered wearable pressure sensing system, as shown in Figure 6d [146]. The tf-TEG consisted of a flexible heat-absorbing film, a hydrogel heat sink, and thin-film thermoelectric materials. The thin-film thermoelectric materials were made of $n$-$Bi_2Te_3$ and $p$-$Sb_2Te_3$, and 25 pairs of the thermoelements were electrically connected in series. The heat absorber was composed of PDMS and boron nitride (BN), which provides high thermal conductivity and low electrical conductivity. The heat sink was formed of hydrogel to maximize the dissipation of heat through the evaporation of water. With conformal attachment onto human skin, the tf-TEG generated a high output voltage of 15.8 mV from body heat at ambient conditions. In addition, the device maintained a high output voltage during 8 h of measurements, and the self-powered pressure system exhibited a high-pressure sensing capability and excellent durability. Wang et al. developed a flexible, transparent, self-powered body movement sensor based on a triboelectric and piezoelectric hybrid generator (TPG) [147]. The TPG consisted of Al-doped ZnO (Al:ZnO) as the transparent electrodes, poly(vinylidene fluoride-trifluoroethylene) (P(VDF-TrFE)) as a piezoelectric functional layer, and micro-structured PDMS as a friction layer, as shown in Figure 6e. The TPG was designed to convert the applied pressure and skin-based friction energy produced from body movements into electricity. A voltage of 26 V and a power density of 751.1 mW/m$^2$ were generated by the TPG under a hand force of 5 N. The TPG attached to human skin produced output voltages of 0.3 V, 1.0 V, 0.6 V, and 0.9 V from the motions of the neck, finger, elbow, and ankle, respectively. In addition to such electricity generation, since different contact and movements of various body parts produced varying output voltage signals, the TPG was also able to serve as a sensor to detect and analyze the movements of the body.

### 5.2. Wireless Data Communication

Wireless data communication is a method of transmitting data from one electronic device to another by networking the electronic devices over radio waves [143]. There is a variety of wireless communication protocols, including Bluetooth, NFC, radio-frequency identification (RFID), and Zigbee. Among them, Bluetooth, BLE, and NFC have been used most extensively in wearable healthcare electronics due to their low energy consumption [6] and high compatibility with various mobile electronic devices, such as smartphones. The details of the wireless communication protocols are summarized in Table 1.

**Table 1.** Summary of characteristics of Bluetooth, BLE, and NFC [6,152,153].

| Protocol | Frequency | Communication Range | Power Consumption | Data Rate | Set-Up Time | Network Type |
|----------|-----------|---------------------|-------------------|-----------|-------------|--------------|
| Bluetooth | 2.4~2.5 GHz | <100 m | <30 mA (read and transmit) | <3 Mb/s | <6 s | WPAN |
| BLE | 2.4~2.5 GHz | <50 m | <15 mA (read and transmit) | <2 Mb/s | <6 ms | WPAN |
| NFC | 13.56 MHz | <0.2 m | 15 mA (read) | 424 kb/s | <0.1 s | Point-to-point |

Bluetooth uses radio frequencies between 2.402 and 2.481 GHz [154]. It can communicate up to ~100 m and transfer larger amounts of data than BLE and NFC. To exchange data between two Bluetooth-enabled devices, they must be paired beforehand. BLE operates at the same radio frequencies of 2.402 and 2.481 GHz, but it is designed to provide low power consumption [155]. The Bluetooth and BLE systems are powered by external power modules, such as batteries that are connected directly to them, which limit the miniaturization and flexibility of the devices incorporating them. Kwon et al. implemented BLE protocols into a wearable EMG sensor patch to enable the real-time wireless transfer of EMG data to an Android tablet, as shown in Figure 6f [148]. The printed circuit board was integrated with miniaturized functional chips, including a BLE/microcontroller, analog-to-digital

converter (ADC), antenna, voltage regulator, and a miniaturized lithium-ion polymer battery. The printed circuit showed comparable Bluetooth performances to that of a rigid circuit up to 15 m. The data transmitted to an Android tablet via Bluetooth were used for wireless human-machine interfaces.

NFC operates at 13.56 MHz and has communication ranges up to 20 cm [156]. Its maximum data transmission rate is 424 kb/s, which is much lower than that of Bluetooth and BLE, but the connection between two NFC systems can be acquired within 0.1 s without prior pairing [157]. Their short communication distance and fast connection provide the NFC systems with high security and low power consumption. In addition, passive NFC systems can be powered wirelessly through inductive coupling with active NFC systems available, for instance, in NFC-enabled mobile phones. Thus, the passive NFC systems only require an antenna circuit and an NFC microchip within the devices for powering, which allows them to be more stretchable, flexible, and miniaturized. Jeong et al. demonstrated e-tattoos capable of wireless power and body signal transmission using NFC protocols, as shown in Figure 6g [17]. The tattoos consisted of an NFC layer, a functional circuit layer, and a passive electrode/sensor layer. The NFC layer was composed of a stretchable antenna with a double-stranded serpentine design, stretchable copper interconnects, a thermistor, and an NFC transponder integrated circuit (IC). The wireless power transmission via NFC enabled the battery-free system, enabling the ultrathin and ultrasoft features of the devices. To power the e-tattoo and record the signals wirelessly, the reader coil that was connected to a USB NFC transponder was placed parallel to it within a distance of 7 cm and the recorded raw signals were saved in the nonvolatile memory of the NFC IC first, and then transmitted to a laptop through a USB NFC transponder wirelessly.

## 6. Wearable Integration

Wearable sensors have been integrated with various platforms, such as contact lenses, tattoos, patches, watches, mouth guards as summarized in Table 2 [1,158,159]. The integrated devices are required to be portable, body-compliant, and able to provide reliable data under dynamic conditions for a prolonged period of time. To fulfill those requirements, all of the components must be configured appropriately and efficiently. This section describes some of the latest research that has demonstrated highly-integrated, wearable, healthcare electronics.

Ku et al. demonstrated a soft, smart contact lens that enables real-time, wireless tacking of cortisol levels in tears using smartphones, as shown in Figure 7a [14]. The lens platform was integrated with a cortisol sensor, transparent antenna, resistor, NFC chip, capacitor, and 3D printed interconnects. The cortisol sensor was fabricated by immobilizing cortisol monoclonal antibodies (C-Mabs) onto the graphene channel of a graphene-field effect transistor (FET). The antigen-antibody interaction between cortisol and the C-Mab induced variations in the drain current, thereby enabling the quantification of the cortisol level. The antenna coil was made of a transparent and stretchable AgNW-AgNF hybrid to ensure that the smart contact lens did not obstruct the wearer's view. The rigid-soft hybrid geometry also used on the substrate enabled efficient arrangement of the components depending on their rigidity and hardness, rendering the entire device stress-resistant. The 3D interconnects were made by direct printing of liquid metal through a nozzle. After the integration of all the components, the resulting flat sample was molded with a PDMS precursor to form the curved shape of a lens. The graphene channel of the sensor was kept uncovered in the process of molding such that tears can make contact with the channel when the resulting smart contact lens was worn on the eye. The wireless power supply from an NFC-enabled smartphone allowed the logic chip to operate and control the operation of the sensor, the recording of data, and the real-time transmission of the data. The data transmitted to a smartphone over NFC was calibrated after which the quantified cortisol levels were displayed on custom software. The resulting data were highly correlated with the actual cortisol concentrations of the tested solutions.

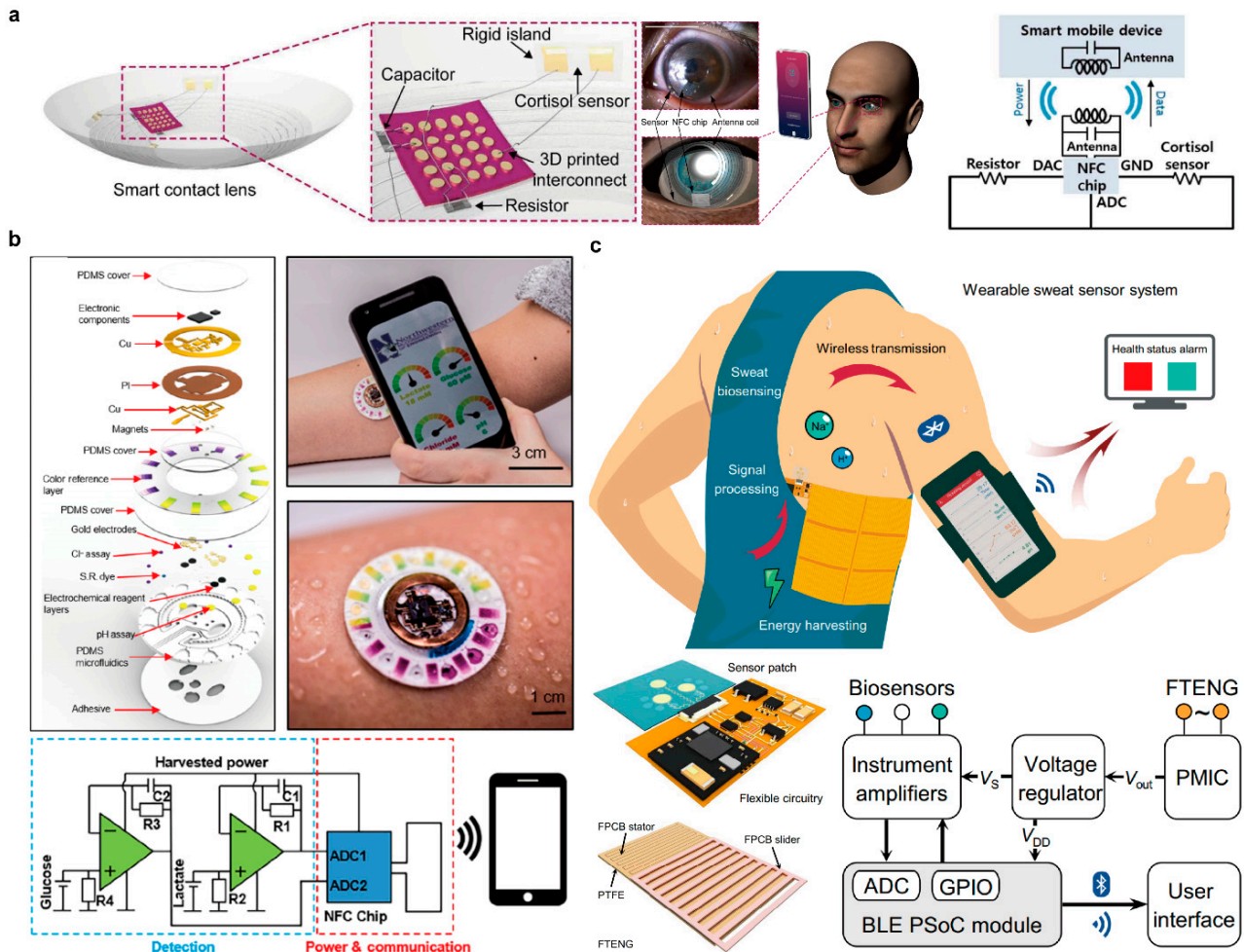

**Figure 7.** (**a**) Smart contact lens for wireless cortisol monitoring integrated with a cortisol sensor, antennas, and wireless communication circuits operated via NFC (reprinted with permission from [14]. Copyright 2020 American Association for the Advancement of Science); (**b**) Wireless skin-interfaced sensor integrated with a microfluidic module, colorimetry sensor, biofuel cell-based electrochemical sensor, battery-free NFC electronics (reprinted with permission from [16]. Copyright 2019 American Association for the Advancement of Science); (**c**) Wireless wearable sweat sensor powered by human motion, equipped with a freestanding triboelectric nanogenerator (reprinted with permission from [160]. Copyright 2020 American Association for the Advancement of Science).

Bandodkar et al. developed an electronic patch capable of battery-free, wireless sensing of lactate, glucose, chloride, pH, and sweat rate/loss, as shown in Figure 7b [16]. The patch consisted of a microfluidic and NFC electronic subsystems that were combined magnetically. The microfluidic subsystem was composed of a microfluidic patch and colorimetric and biofuel cell sensors. Chloride levels and pH were monitored by colorimetric assays, while sweat rates were measured by volumetric analysis. The concentrations of lactose and glucose were measured by the biofuel cell sensor. Notably, the biofuel cell sensor was able to generate electric currents in proportion to the glucose and lactate levels, which eliminates the need for a potentiostat. The NFC electronic subsystem that consisted of NFC chips, amplifiers, antenna, resistors, and capacitors enabled the wireless transmission of the electrochemical sensing data to mobile electronics without a battery. The skin-adhesive membrane that was bonded strongly to the patch enabled robust and conformal attachment of the patch to the skin, even during physical exercise. Since the two subsystems were attached magnetically, they could be separated easily, allowing users to reuse the NFC electronic subsystem and dispose of the microfluidic patch after a single use, which makes the entire system more cost-effective. In rigorous 2-day field studies that involved

multiple 15 to 20 min cycling sessions, the electronic patch showed a high correlation of data acquired by the glucose and lactate sensors within it with corresponding blood levels.

Song et al. proposed a fully self-powered, wearable patch (FWS$^3$) that is capable of the wireless monitoring of key sweat biomarkers including Na$^+$ and pH during exercise, as shown in Figure 7c [160]. The patch was fabricated by integrating a wearable freestanding triboelectric nanogenerator (FTENG), low-power wireless sensor circuitry, and a microfluidic sweat sensor patch on a polyimide-based flexible printed circuit board (FPCB) platform. A flexible biosensor array containing a pH sensor and a Na$^+$ sensor was patterned on a flexible polyethylene terephthalate (PET) substrate. Then, laser patterned microfluidic layers were attached to a PET sensor substrate and Waterproof medical tapes were attached to the microfluidic layers to ensure the conformal lamination of the patch on the skin during exercise, thereby maximizing the efficiency of energy harvesting from human motion. For effective power management, the system utilized a commercial energy harvesting power management integrated circuit (PMIC), low-power instrumentation amplifiers with shutdown modes, and BLE advertisements. The system was designed in a way that the power that was stored in the capacitor was released only when the voltage of the storage capacitors reached a specific threshold. Therefore, when the storage capacitor was fully charged, the capacitor released the energy to power the BLE programmed system on a chip module and the instrumentation amplifiers to record and transmit the signals acquired by the potentiometric sensors over BLE. In human pilot trials, the FWS$^3$ showed a reliable and stable wireless operation during exercise.

## 7. Conclusions and Perspectives

Unlike conventional wearable sensing systems, which generally have bulky, planar, and rigid components, current research has focused on constructing fully-wearable, non-invasive sensing systems. Such systems are aimed at providing improved convenience as well as accurate diagnosis and clinical insights. Remarkable progress has been made in wearable sensing devices based on multi-disciplinary research that included the design of materials, sensors, and wireless technologies as well as their integration. Herein, we have provided a comprehensive review of the progress that has been made concerning the primary factors that are required to construct fully-wearable and integrated sensors. Metal-based, polymeric, and ionic conductors were combined with various structural designs, such as serpentine, wavy, 3D structures to create novel wearable materials. Such materials have been incorporated in the fabrication of wearable sensors to maximize the capabilities of the sensors when subjected to various conditions. Attempts were made to utilize wireless technologies and construct smart integration systems.

Despite such progress, there have been challenges in the operational reliability and durability of current wearable sensors. Developing multiplexed sensing systems could be an option to overcome the reliability issues since such systems allow a comprehensive analysis through the collective input of various parameters. In addition, efforts are needed to develop novel algorithms to properly and efficiently process raw data into vital and useful information. Meanwhile, efforts have been made to find alternative materials and structures to improve the durability of wearable sensors. However, this might result in reduced sensitivity and require complicated fabrication procedures and expensive materials. Therefore, another alternative strategy could be to develop structural designs that enable the simple replacement of components in the wearable sensor. This allows the long-term use of the sensors and benefits their economic perspective. With such improvements, we will get much closer to our ultimate goal of producing completely bio-integrated systems.

**Table 2.** Examples of recently reported wearable sensors.

| Platform | Sensor | Working Principle | Sensing Capability | Power Supply | Data Communication | Refs. |
|---|---|---|---|---|---|---|
| Contact lens | Biochemical | FET-based cortisol sensor, Antigen-antibody interaction | LOD: 10 pg/mL Sensitivity: 1.84 ng/mL per 1% of resistance change. | Inductive coupling | NFC | [14] |
| | Biochemical | Electrochemical glucose sensor, Enzyme-catalyzed | LOD: 12.57 μM, Sensitivity: −22.72%/mM | Inductive coupling | LED display | [15] |
| | Biochemical | Electrochemical glucose sensor | LOD: 4.9 μg/mL | Inductive coupling | RF communication | [161] |
| | Physical | Strain sensor-based intraocular pressure sensor. | LOD: 0.009 mmHg | Inductive coupling | RF communication | [162] |
| Patch | Biochemical, physical | Electrochemical lactate, glucose sensor, colorimetric pH, $Cl^-$ sensor. Microfluidic sweat rate/loss sensor | - | Lactate and glucose in sweat, inductive coupling for electrochemical sensing | NFC | [16] |
| | Biochemical | Microfluidic based sweat sensor | Sensitivity (pH, $Na^+$): 56.28, 58.63 mV/decade | human motion | Bluetooth | [160] |
| | Biochemical | Microfluidic sweat sensor | Sensitivity ($Na^+$): 56 mV/decade | Lithium ion battery | Bluetooth | [163] |
| | Physical | Resistive temperature sensor | Sensitivity 658.5 Ω/°C for 30–42 °C | Lithium ion battery | Bluetooth | [164] |
| tattoo | Biochemical, physical | ECG, Blood oxygen saturation level, and heart rate sensor | - | Inductive coupling | NFC | [17] |
| Mouthguard | Biochemical | Electrochemical $Na^+$ sensor | LOD: 100 μM Sensitivity: 188 ± 12 mV/ decade (gain = 2) | Micro-lithium rechargeable battery | Bluetooth communication | [165] |
| Tooth-mounted patch | Physical | Dielectric sensor | Sensitivity: 0.6 MHz in 1 mg/mL of glucose | Inductive coupling | RF communication | [166] |

**Author Contributions:** S.M.Y., M.K., Y.W.K., H.K., M.J.K. and Y.-G.P. contributed equally to this work. S.M.Y., M.K., Y.W.K., H.K., M.J.K. and Y.-G.P. wrote the paper. J.-U.P. supervised the work. All authors have read and agreed to the published version of the manuscript.

**Funding:** This work was supported by the Ministry of Science & ICT (MSIT) and the Ministry of Trade, Industry and Energy (MOTIE) of Korea through the National Research Foundation (2019R1A2B5B03069358 and 2016R1A5A1009926), the Bio & Medical Technology Development Program (2018M3A9F1021649), the Nano Material Technology Development Program (2016M3A7B4910635), and the Technology Innovation Program (20010366 and 20013621, Center for Super Critical Material Industrial Technology). The authors thank financial support from the Research Program (2019-22-0228) funded by Yonsei University.

**Institutional Review Board Statement:** Not applicable.

**Informed Consent Statement:** Not applicable.

**Conflicts of Interest:** The authors declare no conflict of interest.

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
