# Peer review of "Recent Advances in Wearable Devices for Non-Invasive Sensing"

_applsci, doi:10.3390/app11031235_

Round 1
Reviewer 1 Report
The manuscript entitled Recent advances in wearable devices for non-invasive diagnosis submitted by the group of Authors represents an extensive review on state-of-the art technologies for nonivasive waerable sensing. Thje manuscript is well structured and including materials, structural design, types of sensors, new wireless technologies, wearabel integration chapters and subchapters to cover most of the recent technologies and principles.
It s hard to include all available publications, but Authors should include some recent like:
doi:10.1016/j.snb.2017.02.095 and some new review papers.
L21. Keywords are missing
Title should be replaced "diagnosis = sensing"
Reviewer 2 Report
The manuscript reviews wearable sensors and systems for non-invasive human physiological measurements. It has addressed the most used and novel materials, as well as structures and methods for integration. The document is well structured and covers enough and relevant information.
I have a few comments that will slightly improve the manuscript.
Page 1 line 37, soft conducting materials. The advances has been also made on flexible material as conductor, not only on soft materials. this is mentioned later in page 3 line 78.
page 3 line 92. start by mentioning Ag NPs, but doesn’t mention it is only an example. particles can be of Au, Cu, Ag, etc. As a review, this needs to be stated and provide only examples of the more relevant ones. This is same for each case mentioned, Nano wires,
Page 3 line 94, it will be more interesting to the reader to mention the material of the flexible stretching matrix of the AgNP, as it was done for the other examples.
Page 3 line 103, what is the composition of the material where the AgNF were deposited on or mixed with? Also, introduce nanowire and the mention nano fibres. hey can be used as synonyms, the most common are Nano wires, but only one can be used though al the document.
Page 5 line 166, the sentence is not finished “ studied conductive polymer for flexible [33,49,50].”
Page 9 line 344. Authors mention Physical parameters, which is too generic, and some examples need to be mentioned here.
Page 9 line 373. The authors mention that epidermal devices can detect physical variables of the skin, such as temperature and strain, without any surgical procedure. Please clarify this, as a surgical procedure usually involves cutting a person’s tissue, and I do not know any procedure that does this to detect physical variables.
Reviewer 3 Report
This is a good review, very clear, well-writen, with an appropriate extension and my recomendation is to publish it. The most important aspects concerning wearable devices have been considered. Hot topics are also described and guide the reader to the next challenges.
The only aspect that is hardly mentioned and deserves a section in a review like this has to do with the processing and nanofabrication methodologies for the production of the devices. Many of these methodologies are mentioned in the Materials and Structural Design section. But just that, mentioned. I understand this is a very wide topic and may enlarge this review, but at least a section concerning the main fabrication methodologies employed, perhaps with a table or scheme with advantages and limitations, with references to other wider reviews on this area could also help the reader to put this part of the fabrication methodology of wearable devices in context.
